# High-Fat High-Sugar Diet-Induced Changes in the Lipid Metabolism Are Associated with Mildly Increased COVID-19 Severity and Delayed Recovery in the Syrian Hamster

**DOI:** 10.3390/v13122506

**Published:** 2021-12-14

**Authors:** Julia R. Port, Danielle R. Adney, Benjamin Schwarz, Jonathan E. Schulz, Daniel E. Sturdevant, Brian J. Smith, Victoria A. Avanzato, Myndi G. Holbrook, Jyothi N. Purushotham, Kaitlin A. Stromberg, Ian Leighton, Catharine M. Bosio, Carl Shaia, Vincent J. Munster

**Affiliations:** 1Laboratory of Virology, National Institute of Allergy and Infectious Diseases, National Institutes of Health, Hamilton, MT 59840, USA; julia.port@nih.gov (J.R.P.); dadney@lovelacebiomedical.org (D.R.A.); jonathan.schulz@nih.gov (J.E.S.); victoria.avanzato@nih.gov (V.A.A.); myndi.holbrook@nih.gov (M.G.H.); jyothi.purushotham@nih.gov (J.N.P.); 2Laboratory of Bacteriology, National Institute of Allergy and Infectious Diseases, National Institutes of Health, Hamilton, MT 59840, USA; benjamin.schwarz@nih.gov (B.S.); kaitlin.stromberg@nih.gov (K.A.S.); ian.leighton@nih.gov (I.L.); bosioc@niaid.nih.gov (C.M.B.); 3Genomics Unit, Research Technologies Branch, National Institute of Allergy and Infectious Diseases, National Institutes of Health, Hamilton, MT 59840, USA; dsturdevant@niaid.nih.gov; 4Rocky Mountain Veterinary Branch, Division of Intramural Research, National Institutes of Health, Hamilton, MT 59840, USA; brian.smith2@nih.gov (B.J.S.); carl.shaia@nih.gov (C.S.); 5Division of Structural Biology, Wellcome Centre for Human Genetics, University of Oxford, Oxford OX3 7BN, UK; 6The Jenner Institute, Nuffield Department of Clinical Medicine, University of Oxford, Oxford OX3 7BN, UK

**Keywords:** Syrian hamster, SARS-CoV-2, obesity, pathogenesis, lipid metabolism

## Abstract

Pre-existing comorbidities such as obesity or metabolic diseases can adversely affect the clinical outcome of COVID-19. Chronic metabolic disorders are globally on the rise and often a consequence of an unhealthy diet, referred to as a Western Diet. For the first time in the Syrian hamster model, we demonstrate the detrimental impact of a continuous high-fat high-sugar diet on COVID-19 outcome. We observed increased weight loss and lung pathology, such as exudate, vasculitis, hemorrhage, fibrin, and edema, delayed viral clearance and functional lung recovery, and prolonged viral shedding. This was accompanied by an altered, but not significantly different, systemic IL-10 and IL-6 profile, as well as a dysregulated serum lipid response dominated by polyunsaturated fatty acid-containing phosphatidylethanolamine, partially recapitulating cytokine and lipid responses associated with severe human COVID-19. Our data support the hamster model for testing restrictive or targeted diets and immunomodulatory therapies to mediate the adverse effects of metabolic disease on COVID-19.

## 1. Introduction

Severe acute respiratory syndrome coronavirus-2 (SARS-CoV-2) is the etiological agent of coronavirus disease (COVID)-19 and can cause asymptomatic to severe lower respiratory tract infections in humans [1,2]. Pre-existing comorbidities such as immunosuppression, obesity, diabetes, or chronic lung disease can adversely affect the clinical outcome [3,4,5,6]. Of these, obesity and metabolic disorders are global pandemics of rising concern [7,8,9]. The underlying disease is driven mainly by changes in the global food system, which is producing more processed, affordable, and effectively marketed food than ever before. This diet, rich in saturated fats and refined sugars, is referred to as a Western Diet [10]. Long-term consumption of a Western Diet may result in chronic activation of the immune system, impairing both innate and adaptive responses [11,12,13]. The Western Diet has been associated with non-alcoholic steatohepatitis (NASH) and non-alcoholic fatty liver disease (NAFLD). These disease syndromes predispose individuals to multiple comorbidities that can include cirrhosis and liver failure. The relative risk of hospitalization and severe COVID-19 outcome are significantly increased for patients afflicted by these comorbidities [3]. This has resulted in disproportionately worse outcomes in US ethnic and racial minorities, where prevalence and incidence of metabolic disorders are increased [14].

It is currently unclear how certain comorbidities may determine disease manifestation of COVID-19. Different studies have demonstrated that the Syrian hamster model is suitable to model aspects of obesity and diabetes and for studying lipid metabolism [15,16]. In healthy hamsters, SARS-CoV-2 infection is associated with mild to moderate clinical disease [17,18,19]. However, no studies have investigated COVID-19 in hamsters with comorbidities. Here, we show in a Syrian hamster model how a continuous high-fat high-sugar (HFHS) diet changed the metabolomic state in the Syrian hamster and the resulting consequences on viral replication dynamics, immune protection, and disease severity after infection with SARS-CoV-2.

## 2. Materials and Methods

### 2.1. Ethics Statement

Approval of animal experiments was obtained from the Institutional Animal Care and Use Committee of the Rocky Mountain Laboratories. Performance of experiments was done following the guidelines and basic principles in the United States Public Health Service Policy on Humane Care and Use of Laboratory Animals and the Guide for the Care and Use of Laboratory Animals. Work with infectious SARS-CoV-2 strains under BSL3 conditions was approved by the Institutional Biosafety Committee (IBC). Inactivation and removal of samples from high containment was performed per IBC-approved standard operating procedures [20].

### 2.2. Virus and Cells

SARS-CoV-2 strain nCoV-WA1-2020 (MN985325.1) was provided by CDC, Atlanta, USA. Virus propagation was performed in VeroE6 cells in DMEM supplemented with 2% fetal bovine serum (FBS), 2 mM L-glutamine, 100 U/mL penicillin, and 100 μg/mL streptomycin. VeroE6 cells were maintained in DMEM supplemented with 10% FBS, 2 mM L-glutamine, 100 U/mL penicillin, and 100 μg/mL streptomycin D10. Virus stock (passage 4) was 100% identical to the initial sequence (MN985325.1) and no contaminants were detected.

### 2.3. High-Fat High-Sugar Diet

Four to six-week-old male Syrian Golden hamsters (ENVIGO, *n* = 70 total) were randomly assigned to either regular rodent chow (Teklad Global 16% Protein Rodent Diet, Envigo, Indianapolis, IN, USA) or a HFHS diet for 16 weeks (Purina Chow #5001 with 11.5% Corn Oil, 11.5% Coconut Oil, 0.5% Cholesterol, 0.25% Deoxycholic Acid, and 10% Fructose: Dyets Inc., Dyet#615088, Bethlehem, PA, USA). Pre-challenge oral glucose tests were performed on all animals. Five animals from each diet group were euthanized after the 16 wks for collection of pre-challenge tissue samples and weights. For each diet group, 5 animals were randomly designated for flexiVent calibration and excluded from further analysis. Three animals in the HFHS regimen were euthanized throughout the 16-week diet regimen due to secondary morbidities and were not included in analyses. Pre-challenge, an additional 5 animals in the RD group and additional 8 animals in the HFHS group were excluded from the study due to experimental reasons, and one animal in the HFHS group due to secondary morbidities.

### 2.4. Assessment of Glucose Tolerance

An oral glucose tolerance test (OGTT) was performed after 16 weeks of diet manipulation [15]. Hamsters were fasted for 16 h overnight preceding the OGTT. An oral glucose load (2 g/kg glucose) was administered. Blood samples were collected from the retroorbital sinus using capillary tube at 0-, 30-, 60-, and 120-min post glucose administration. Blood glucose was measured using the AlphaTRAK blood glucose monitoring system (Zoetis, Florham Park, NJ, USA), calibrated for cats. Serum was separated and used for measurement of insulin. Insulin was measured using the rat/mouse insulin ELISA kit from Millipore (Burlington, MA, USA) (EZRMI-13K), according to the manufacturer’s instructions [21].

### 2.5. Lipidomics

Blood lipids were assessed for a subset of animals (*n* = 8–10) after 16 weeks of diet. A total of 200 µL blood was collected and were measured using the Piccolo^®^ Lipid Panel Plus for humans (Abaxis, Union City, CA, USA) according to the manufacturer’s instruction.

### 2.6. Next-Generation Sequencing of Liver mRNA

Frozen tissues were pulverized in 1 mL of Trizol (ThermoFisher Scientific, Waltham, MA, USA), 200 µL of 1-Bromo-3-chloropropane (MilliporeSigma, St. Louis, MO, USA) was added, samples mixed, and centrifuged at 16,000× *g* for 15 min at 4 °C. RNA containing aqueous phase of 600 µL was collected from each sample and passed through Qiashredder column (Qiagen, Hilden, Germany) at 21,000× *g* for 2 min to homogenize any remaining genomic DNA in the aqueous phase. Aqueous phase was combined with 600 µL of RLT lysis buffer (Qiagen, Valencia, CA, USA) with 1% beta mercaptoethanol (MilliporeSigma, St. Louis, MO, USA) and RNA was extracted using Qiagen AllPrep DNA/RNA 96-well system. An additional on-column DNase-1 treatment was performed during RNA extraction. RNA was quantitated by spectrophotometry and yield ranged from 0.4 to 17.8 µg. One hundred nanograms of RNA was used as input for rRNA depletion and NGS library preparation following the Illumina Stranded Total RNA Prep Ligation with Ribo-Zero Plus workflow (Illumina, San Diego, CA, USA). The NGS libraries were prepared, amplified for 13 cycles, AMPureXP bead (Beckman Coulter, Pasadena, CA, USA) purified using 0.95X beads, assessed on a BioAnalyzer DNA1000 chip (Agilent Technologies, Santa Clara, CA, USA) and quantified using the Kapa Quantification Kit for Illumina Sequencing (Roche, Basel, Switzerland). Amplified libraries were pooled at equal molar amounts and sequenced on a NextSeq (Illumina, San Diego, CA, USA) using two High Output 150 cycle chemistry kits. Raw fastq reads were trimmed of Illumina adapter sequences using cutadapt version 1.12 and then trimmed and filtered for quality using the FASTX-Toolkit (Hannon Lab, University of Cambridge, Cambridge, UK). Remaining reads were aligned to the *Mesocricetus auratus* genome assembly version 1.0 using Hisat2 [22]. Reads mapping to genes were counted using htseq-count [23]. Differential expression analysis was performed using the Bioconductor package DESeq2 [24]. Pathway analysis was performed using Ingenuity Pathway Analysis (QIAGEN) and gene clustering was performed using Partek Genomics Suite (Partek Inc., Chesterfield, MO, USA). Samples with too low quality were removed from the analysis (Appendix A).

### 2.7. Next-Generation Sequencing of Virus

For sequencing from viral stocks, sequencing libraries were prepared using Stranded Total RNA Prep Ligation with Ribo-Zero Plus kit per manufacturer’s protocol (Illumina, San Diego, CA, USA) and sequenced on an Illumina MiSeq at 2 × 150 base pair reads. For sequencing from swab and lung tissue, total RNA was depleted of ribosomal RNA using the Ribo-Zero Gold rRNA Removal kit (Illumina). Sequencing libraries were constructed using the KAPA RNA HyperPrep kit following manufacturer’s protocol (Roche Sequencing Solutions). To enrich for SARS-CoV-2 sequence, libraries were hybridized to myBaits Expert Virus biotinylated oligonucleotide baits following the manufacturer’s manual, version 4.01 (Arbor Biosciences, Ann Arbor, MI, USA). Enriched libraries were sequenced on the Illumina MiSeq instrument as paired-end 2 × 150 base pair reads. Raw fastq reads were trimmed of Illumina adapter sequences using cutadapt version 1.1227 and then trimmed and filtered for quality using the FASTX-Toolkit (Hannon Lab, CSHL, University of Cambridge, Cambridge, UK). Remaining reads were mapped to the SARS-CoV-2 2019-nCoV/USA-WA1/2020 genome (MN985325.1) using Bowtie2 version 2.2.928 with parameters --local --no-mixed -× 1500. PCR duplicates were removed using picard MarkDuplicates (Broad Institute, Cambridge MA, USA) and variants were called using GATK HaplotypeCaller version 4.1.2.029 with parameter -ploidy 2. Variants were filtered for QUAL > 500 and DP > 20 using bcftools.

### 2.8. Inoculation Experiments

After 16 weeks, animals were then inoculated intranasally (I.N.) under isoflurane anesthesia. I.N. inoculation was performed with 40 µL sterile Dulbecco’s Modified Eagle Medium (DMEM) containing 8 × 10^4^ TCID50 SARS-CoV-2. A subset of animals (*n* = 4–10) were euthanized, and serum and tissues were collected at pre-challenge (0 DPI), 4, 7, 14, and 21 DPI. Hamsters were weighted daily, and oropharyngeal swabs (21 DPI animals only) were taken daily until day 7 and then thrice a week. Swabs were collected in 1 mL DMEM with 200 U/mL penicillin and 200 µg/mL streptomycin. Hamsters were observed daily for clinical signs of disease.

### 2.9. Lung Function Analyses

Lung function assessment was performed on pre-challenge, 7, 14, and 21 DPI. Hamsters were anesthetized with a combination of inhalant isoflurane and ketamine/xylazine intraperitoneally. After animals reached a surgical plane of anesthesia a terminal tracheostomy was performed as previously described [25]. Briefly, a cannula was introduced into the trachea, secured with suture, and the animal underwent the forced oscillation technique (FOT) using a flexiVent (SCIREQ, Inc. Montreal, QC Canada). Animals were kept at a consistent surgical plane of anesthesia to the point of not resisting the FOT procedure. Animals were immediately euthanized while deeply anesthetized after FOT was completed; the surgical procedure was terminal.

### 2.10. Histopathology and Immunohistochemistry

Necropsies and tissue sampling were performed according to IBC-approved protocols. Tissues were fixed for a minimum of 7 days in 10% neutral buffered formalin with 2 changes. Tissues were placed in cassettes and processed with a Sakura VIP-6 Tissue Tek, on a 12 h automated schedule, using a graded series of ethanol, xylene, and ParaPlast Extra. Prior to staining, embedded tissues were sectioned at 5 µm and dried overnight at 42 °C. Using GenScript U864YFA140-4/CB2093 NP-1 (1:1000) specific anti-CoV immunoreactivity, CD3 (Predilute) (Roche Tissue Diagnostics #790-4341), and PAX5 (1:500) (Novus Biologicals #NBP2-38790, Littleton, CO, USA) were detected using the Vector Laboratories ImPress VR anti-rabbit IgG polymer (# MP-6401) as the secondary antibody. Iba-1 (1:500) (abcam #ab5076) was detected using Roche Tissue Diagnostics OmniMap anti-goat multimer (#760-4647) as the secondary antibody. The tissues were stained using the Discovery Ultra automated stainer (Ventana Medical Systems, Tucson, AZ, USA) with a ChromoMap DAB kit Roche Tissue Diagnostics (#760-159). Histopathology was assessed by a board-certified veterinary pathologist using criteria as previously applied to the Syrian hamster SARS COV-2 model.

### 2.11. Morphometric Analysis

IHC stained tissue slides were scanned with an Aperio ScanScope XT (Aperio Technologies, Inc., Cumming, GA, USA, 30041) and analyzed using the ImageScope Positive Pixel Count algorithm (version 9.1). The default parameters of the Positive Pixel Count (hue of 0.1 and width of 0.5) detected antigen adequately.

### 2.12. Viral RNA Detection

Swabs from hamsters were collected as described above. Then, 140 µL was utilized for RNA extraction using the QIAamp Viral RNA Kit (Qiagen, Hilden Germany) using QIAcube HT automated system (Qiagen) according to the manufacturer’s instructions with an elution volume of 150 µL. Sub-genomic (sg) viral RNA and genomic (g) was detected by qRT-PCR [26]. Then, 5 μL RNA was tested with TaqMan™ Fast Virus One-Step Master Mix (Applied Biosystems, Foster City, CA, USA) using QuantStudio 6 Flex Real-Time PCR System (Applied Biosystems) according to instructions of the manufacturer. Ten-fold dilutions of SARS-CoV-2 standards with known copy numbers were used to construct a standard curve and calculate copy numbers/mL.

### 2.13. Viral Titration

Viable virus in tissue samples was determined as previously described [27]. In brief, lung tissue samples were weighed, then homogenized in 1 mL of DMEM2. VeroE6 cells were inoculated with 10-fold serial dilutions of tissue homogenate, spun at 1000 rpm for 1 h at 37 °C, the first dilutions washed with PBS and with DMEM2. Cells were incubated with tissue homogenate for 6 days at 37 °C, 5% CO_2_, then scored for cytopathic effect. TCID50 was calculated by the method of Spearman–Karber and adjusted for tissue weight.

### 2.14. Serology

Serum samples were inactivated with γ-irradiation (2 mRad) and analyzed as previously described [28]. In brief, maxisorp plates (Nunc) were coated with 50 ng spike protein (generated in-house) per well and incubated overnight at 4 °C. After blocking with casein in phosphate buffered saline (PBS) (ThermoFisher) for 1 h at room temperature (RT), serially diluted 2-fold serum samples (duplicate, in blocking buffer) were incubated for 1 h at RT. Spike-specific antibodies were detected with goat anti-hamster IgG Fc (horseradish peroxidase (HRP)-conjugated, Abcam) for 1 hr at RT and visualized with KPL TMB 2-component peroxidase substrate kit (SeraCare, 5120-0047, Milford, MA, USA). The reaction was stopped with KPL stop solution (Seracare) and read at 450 nm. Plates were washed 3 to 5× with PBS-T (0.1% Tween) for each wash. The threshold for positivity was calculated as the average plus 3× the standard deviation of negative control hamster sera.

### 2.15. Cytokine Analysis

Cytokine concentrations were determined using a commercial hamster ELISA kit for TNF-α, IFN-γ, IL-6, IL-4, and IL-10 available at antibodies.com, according to the manufacturer’s instructions (antibodies.com; A74292, A74590, A74291, A74027, A75096). Samples were pre-diluted 1:10.

### 2.16. Serum Lipid Analysis

For abundance analysis of serum lipids signals were filtered using a 50% miss value cut off and applying a raw intensity cutoff appropriate to the noise level of each class of lipids. Signals were then normalized to internal deuterated SPLASH^®^ LIPIDOMIX^®^ Mass Spec Standard (Avanti Polar Lipids, Alabaster, AL, USA). For compositional analysis of the serum, bulk lipid datasets were further filtered using a 30% QC coefficient of variance cut off prior to normalizing by the total signal sum. All univariate and multivariate analysis was performed using GraphPad Prism or MarkerView (AB Sciex, Redwood City, CA, USA). All parallel univariate analysis was subjected to a Benjamini–Hochberg correction using a false discovery rate of 15%.

### 2.17. Statistical Analysis

All graphs were designed in GraphPad Prism software (version 8.0.1; GraphPad Software). Significance tests were performed as indicated where appropriate. Statistical significance levels were determined as follows: ns = *p* > 0.05; * = *p* ≤ 0.05; ** = *p* ≤ 0.01; *** = *p* ≤ 0.001; **** = *p* ≤ 0.0001.

## 3. Results

### 3.1. High-Fat and High-Sugar Diet Induces Metabolic Changes Characterized by Increased Early Weight Gain and Glucose Tolerance

We investigated the impact of a consistent high-fat and high-sugar (HFHS) diet on the Syrian hamster. Either a regular rodent (RD) diet or a high-calorimetric HFHS diet was given to male Syrian hamsters (4–6-week-old) for 16 weeks ad libitum (*n* = 35, respectively). Weight gain of juvenile hamsters was monitored weekly. Initially, animals on the HFHS diet gained weight faster than animals on the regular diet, although this was a transient difference. Difference in median weights was significant from the 2nd week onwards until week 10 (Figure 1A, *n* = 35, ordinary two-way ANOVA, followed by Sidak’s multiple comparisons test, *p* = 0.001, *p* = <0.001, *p* = <0.001, *p* = <0.001, *p* = <0.001, *p* = <0.001, *p* = <0.001, *p* = 0.0011, *p* = <0.001). After week 10, weight gain either plateaued or decreased in the HFHS group (median = 165 g), while in the regular diet group weight increased until week 12 (median = 160 g), at which point the median weight between groups showed no significant difference. We observed morbidity (4/35 = 11%) in the HFHS group, which was absent in the RD group.

To assess the levels of glucose-associated symptoms triggered by a HFHS diet we conducted an oral glucose tolerance test (OGTT). No difference in fasting blood glucose levels between diet groups was observed (*n* = 30 (RD)/29 (HFHS), median = 150/147 mg/dL). However, HFHS animals demonstrated impaired glucose intolerance upon application of an oral glucose dose; blood glucose levels 30, 60, and 120 min after oral application were significantly increased compared to RD animals (Figure 1B, *n* = 30 (RD)/29 (HFHS), 30 min median = 265/313 mg/dL and 60 min median = 290/347 mg/dL, ordinary two-way ANOVA, followed by Sidak’s multiple comparisons test, *p* = 0.0004, *p* = 0.0009, respectively). We compared the insulin response after application of oral glucose load and found no difference between the diet regimens. The insulin resistance index (fasting glucose level (mmol/L) x fasting insulin level (mIU/L) showed no significant differences (Figure 1C, *n* = 30 (RD)/29 (HFHS), Mann–Whitney test, *p* = 0.6871) [4,29]. Five animals were euthanized pre-challenge in order to assess diet induced pathology. There was no difference in body fat-to-weight ratio (Figure 1D, *n* = 5, median = 1.905 (RD)/2.117 (HFHS) Fat:Bodyweight ratio (mg/g), Mann–Whitney test, *p* > 0.9999).

### 3.2. High-Fat and High-Sugar Diet Induces Liver Damage and Systemic Hyperlipidemia

We investigated the changes in lipid metabolism through a blood lipid biochemistry panel (Appendix A). Due to increased levels of fat in the samples collected from HFHS animals, HDL and LDL could not be assessed, as some values were too high for the instrument to read. Total cholesterol was increased in the HFHS group (Figure 1E, *n* = 10 (RD)/7 (HFHS), median = 67.6/380 mg/dL). The median (146 U/L) alanine aminotransferase (ALT), an indication of hepatocellular injury without overt cholestasis, values in the HFHS animals were above the upper limit of previously established reference ranges [30]. To understand which lipids were circulating in serum, we analyzed serum by liquid chromatography tandem mass spectrometry (LC-MS/MS). Aggregate signals across all lipid classes assayed in the HFHS animals compared to RD were increased, comprising phospholipids, cholesterol esters, sphingolipids, neutral lipids, lysophospholipids, and free fatty acids (Figure 1F, *n* = 5(RD)/4 (HFHS), Mann–Whitney test, *p* = 0.0159, *p* = 0.0635, *p* = 0.0159, *p* = 0.0317, *p* = 0.0653, *p* = 0.0317, respectively). Hence, we further assessed changes in the liver through gross and histologic pathology. Gross pathology of livers differed substantially. Livers from animals on the HFHS diet were diffusely pale, friable, and sections floated in formalin while RD hamster livers appeared grossly normal. Histologically, hepatocytes were expanded by micro and macrovesicles in HFHS animals, while hepatocytes in RD animals appeared normal (Figure 2A–F).

To further characterize the effect of the HFHS diet regimen on the liver, we evaluated global changes in the gene expression after 16 weeks. Principal components analysis of the complete gene expression profile revealed expected grouping with each diet regimen group containing their associated replicates (Appendix A, *n* = 5 (RD), 4 HFHS). In total, 2114 genes were significantly, differentially expressed (*p* < 0.05 and >2 fold) in the liver. To assess the enrichment of these differential genes, they were imported into Ingenuity Pathway Analysis (IPA) software. The results show that in the comparison of HFHS to RD animals 124 canonical pathways were significantly enriched and 200 downstream effects were predicted on biological processes and disease or toxicological function (*p*-value < 0.05, z-score ≤ −2 or ≥ 2): amongst which were cell recruitment, inflammation, activation, and immune-associated pathways (Figure 2G, Appendix A shows all significant predicted downstream effects). Interestingly, we also observed a pathway activation pattern reminiscent of NAFLD TNF-driven inflammation, (Figure 2H).

Together, these data suggest that HFHS diet induced drastic changes in glucose uptake and lipid metabolism, characterized by systemic dyslipidemia and gross changes in liver pathology. This translated into increased inflammation and a gene expression profile in the liver reminiscent of fatty liver disease.

### 3.3. High-Fat and High-Sugar Diet Exacerbated Disease Severity after SARS-CoV-2 Infection

We challenged hamsters (RD: *n* = 20, HFHS = 13 (group size adjusted for the HFHS group due to the morbidity of the model pre-challenge)) with 8 × 10^4^ TCID50 SARS-CoV-2 via the intranasal route. Animals were euthanized at 7 days-post inoculation (DPI) (RD: *n* = 10, HFHS = 4), at 14 DPI (RD: *n* = 5, HFHS = 4) or monitored until 21 DPI (RD: *n* = 5, HFHS) = 5. We observed marginally more severe morbidity in the HFHS group, in which two animals reached euthanasia criteria (>20% relative body weight loss) at 8 and 9 DPI, respectively (Figure 3A). While the HFHS animals demonstrated non-infection associated morbidity, the timing and symptoms associated with these fatalities suggest that they were caused by the infection. In the RD group, a median peak weight loss was observed at 6 DPI (~7% relative body weight), after which animals recovered and returned to pre-challenge weights by 14 DPI. Weight in HFHS animals was significantly decreased after 3 DPI and negative area under the curve (AUC) analysis between 1 and 14 DPI revealed significant difference (Figure 3B, *n* = 10 (RD)/7 (HFHS), Mann–Whitney test, *p* = 0.0002). In the HFHS group median peak weight loss was reached at 8 DPI (~16% relative body weight) and no animal recovered pre-challenge weights until the end of the study at 21 DPI.

To better understand the clinical impact of a HFHS diet on SARS-CoV-2 infection, the respiratory function of the hamsters was evaluated. We performed forced oscillation tests on mechanically ventilated hamsters pre-challenge, and on 7, 14, and 21 DPI. No significant differences in pulmonary function were detected between the RD and HFHS groups at any time point.

Pulmonary function after SARS-CoV-2 infection has not been assessed in the Syrian hamster yet, so we combined the groups to evaluate changes over the course of infection. Inspiratory capacity was significantly decreased in 7 DPI as compared to pre-challenge (Figure 3C, baseline: *n* = 5 (RD)/3 (HFHS) and 7 DPI: *n* = 5 (RD)/4 (HFHS), baseline median = 4.345/4.032 and 7 DPI median = 3.195/3.464 mL, ordinary two-way ANOVA, followed by Tukey’s multiple comparisons test, *p* = 0.0107). Elastance of the respiratory system was significantly increased at 7 DPI (baseline median = 2.68/3.032 and 7 DPI median = 4.138/3.852 cmH2O/mL, *p* = 0.0022), as was tissue elastance (baseline median = 2.514/2.450 and 7 DPI median = 3.021/3217 cmH2O/mL, *p* = 0.0040). The resistance of the airway not associated with gas exchange (Newtonian resistance) was not significantly different at any time point; however, total resistance was significantly increased in 7 DPI as compared to pre-challenge (baseline median = 0.151/0.167 and 7 DPI median = 0.181/0.205 cmH2O.s/mL, *p* = 0.034). Changes in peripheral resistance were also detected by an increase in tissue damping at 7 DPI as compared to pre-challenge animals, which reflects how oscillatory energy is dispersed or retained within parenchymal tissue (baseline median = 0.564/0.623 and 7 DPI median = 0.695/0.720 cmH2O/mL, *p* = 0.0158). Recovery to pre-challenge was observed for all parameters by 14 DPI. Together, these changes in respiratory function led to an overall decrease in shape parameter k, which reflects the curvature of the pressure–volume curve, on 7 DPI (Figure 3D, baseline median = 0.193/0.180 and 7 DPI median = 0.168/0.158/cmH20, ordinary two-way ANOVA, followed by Sidak’s multiple comparisons test, *p* = 0.0001). While not significant, a slower recovery to pre-challenge values for resistance and tissue damping was observed in the HFHS group. This could indicate that functional lung recovery in this group was slower.

### 3.4. High-Fat and High-Sugar Diet Is Associated with Exudate, Vasculitis, Inflammation of the Epithelia and Hemorrhage, Fibrin and Edema, and Decreased Viral Clearance

Next, we assessed the pathology in the lungs at necropsy, 7 DPI. Grossly, lungs displayed lesions with multifocal dark red foci visible on the surface of the lobes (Figure 4A–J). Across groups the 7 DPI lungs were more turgid, failed to collapse, and had increased lung weights as compared to pre-challenge lungs (Appendix A). Lung weight recovery appeared slower in HFHS animals. Histopathologically, only a subset of RD animals demonstrated increased lung damage (*n* = 5/10, >50% lung tissue affected). At 7 DPI, foci were multifocal and adjacent to bronchi and blood vessels as well as peripherally along the sub pleural margin. Overall, no significant difference was seen between the cumulative pathological score between diet groups. However, three out of four animals demonstrated lesions in >50% of tissue (Figure 4K and Appendix A). In HFHS animals, foci were multifocal but less clearly delineated due to hemorrhage, edema, and fibrin. Interstitial pneumonia was characterized by thickened septa due to inflammatory cells, fibrin and edema and lined by hyperplastic type II pneumocytes. Alveoli were filled with inflammatory cells, edema and organizing fibrin. The two HFHS animals which were euthanized at day 8/9 due to severe disease and weight loss (>20%) both showed pneumonia, hemorrhage, edema, and inflammation (Appendix A).

At 14 DPI, thickened septa, presumably from interstitial fibrosis with alveolar bronchiolization, were observed in lungs from RD animals (*n* = 2) (Appendix A). In contrast, HFHS animals at 14 DPI had less septal thickening and more septal, alveolar, and perivascular inflammation (*n* = 2). At 21 DPI four out of five of the RD animals and three out of three of the HFHS animals had thickened alveolar septa with alveolar bronchiolization (Appendix A).

Immunohistochemistry staining for SARS-CoV-2 antigen was increased at 7 DPI in lungs of HFHS animals compared to RD animals (median = 2.71 (RD)/5.043 (HFHS), *n* = 10/4) (Figure 4E,J,L). To confirm this finding, we compared genomic RNA, subgenomic (sg) RNA (surrogate for replicating virus), and infectious viral particles isolated from lungs at 7 DPI. Levels of gRNA and sgRNA in the lungs of HFHS animals at 7 DPI were significantly increased as compared to RD animals. Additionally, no infectious virus could be isolated from a subset of RD animals and overall, significantly more infectious virus could be isolated in HFHS animals (Figure 4M–O; RD: *n* = 10, HFHS: *n* = 4, gRNA median = 6.935/8.513 copies/g lung (log10), sgRNA median = 5.639/7.896 copies/g lung (log10) and infectious virus median = 1.63/3.703 TCID50/g (log10), Mann–Whitney test, *p* = 0.0240, *p* = 0.0240 and *p* = 0.0120, respectively).

To better understand if the HFHS diet contributed to changes in viral replication kinetics in the upper respiratory tract, swabs from the oropharynx were analyzed for the presence of sgRNA. Respiratory shedding in both groups peaked at 2 DPI. Shedding in HFHS animals was constantly high up until 10 DPI, while shedding began decreasing in RD animals after 6 DPI. To compare the overall shedding burden, we performed an area under the curve (AUC) analysis for both groups depicting the cumulative shedding. HFHS animals presented significantly higher cumulative shedding (Figure 4P,Q, *n* = 5 (RD)/3 (HFHS), median 41.48/44.44 AUC (log10), Mann–Whitney test, *p* = 0.0357).

### 3.5. Immune Infiltration in the Lung during the Acute-Phase of Infection and Humoral Immunity Are Not Significantly Affected by High-Fat High-Sugar Diet

Using immunohistochemistry, we investigated the infiltration of macrophages (IBA 1 staining), T-cells (CD3 staining), and B-cells (Pax 5 staining) over the course of infection (Figure 5). Macrophages were detected throughout all sections but were increased in 7 and 14 DPI samples in pneumonic areas irrespective of diet regimen. In addition, T lymphocytes were increased in 7 and 14 DPI samples in pneumonic areas. No increase in B cells was observed. To quantify the influx of macrophages and T cells we used morphometric analysis (Appendix A). No significant difference was seen between the RD and HFHS groups. Both macrophages and T cells increased in numbers at 7 DPI as compared to pre-challenge conditions for both groups. (Figure 6A,B, pre-challenge: *n* = (RD)/2 (HFHS) and 7 DPI: *n* = 10 (RD)/4 (HFHS), median macrophages = (3.075/3.530 (pre-challenge))/(13.630/10.480 (7 DPI)) % reactivity and median T cells = (4.515/4.125 (pre-challenge))/(11.340/11.255 (7 DPI)) % reactivity, ordinary two-way ANOVA, followed by Sidak’s multiple comparisons test, *p* = 0.1007/0.3564 and *p* = 0.0001/0.0001, respectively).

The humoral response to SARS-CoV-2 was not significantly impacted by diet regimen. Animals seroconverted at 7 DPI, as measured by anti-spike IgG ELISA (Figure 6C, 7 DPI: *n* = 10 (RD)/4 (HFHS), 14 DPI: *n* = 5 (RD)/4 (HFHS), 21 DPI: *n* = 5 (RD)/3 (HFHS), ordinary two-way ANOVA, followed by Tukey’s multiple comparisons test, *p* = 0.8573, *p* = 0.8203 and *p* = 0.5468, respectively). Neutralization of virus by sera collected at 14 and 21 DPI was compared to assess potential differences in affinity maturation and no significant difference was found (Figure 6D, 14 DPI: *n* = 5 (RD)/4 (HFHS), 21 DPI: *n* = 5 (RD)/3 (HFHS), 14 DPI median = 120/80 and 21 DPI median = 120/120 reciprocal titer, ordinary two-way ANOVA, followed by Tukey’s multiple comparisons test, *p* = 0.5535 and *p* = 0.4688, respectively).

### 3.6. Prolonged SARS-CoV-2 Shedding, Systemic Immune and Metabolomic Dysregulation after High-Fat High-Sugar Diet

The cytokine kinetics were analyzed in serum throughout the course of infection by ELISA. Serum samples were collected pre-challenge (0 DPI), on 7, 14, and 21 DPI (Figure 6E). Pro-inflammatory tumor necrosis factor (TNF)-α, interleukin (IL)-6, antiviral interferon (IFN)-γ, and IL-10 did not significantly differ between diet regimens pre-challenge. After infection, RD animals mounted a significant IFN-γ response which lasted into recovery (14 and 21 DPI), while no response was seen in HFHS animals (RD: *n* = 5/10, HFHS: *n* = 4, pre-challenge median = 629/618, 7 DPI median = 737.85/550.6, 14 DPI median = 702.3/623.55, 21 DPI median = 1042.3/609.8 pg/mL, ordinary two-way ANOVA, followed by Sidak’s multiple comparisons test, pre-challenge: *p* = 0.58157, 7 DPI: *p* = 0.0090, 14 DPI: *p* = 0.7373, 21 DPI *p* < 0.0001). In contrast, serum IL-6 was increased in HFHS animals compared to RD animals at 7 DPI (median = 2795.5 (RD)/2859.2 (HFHS) pg/mL), but this was found to be not significant. IL-10 levels were equally increased in some HFHS animals during the acute phase and remained elevated at 14 DPI (RD: *n* = 5/10, HFHS: *n* = 4, pre-challenge median = 1894.6/2131.5, 7 DPI median = 2071.75/2773.95, 14 DPI median = 1768.5/2354.35, 21 DPI median = 1733.7/2407.6 pg/mL ordinary two-way ANOVA, followed by Sidak’s multiple comparisons test, pre-challenge: *p* = 0.9933, 7 DPI: *p* = 0.0548, 14 DPI: *p* = 0.1408, 21 DPI *p* = 1259). This was found not to be significant. TNF-α serum levels demonstrated an ambivalent pattern.

To examine compositional changes in the circulating lipidome over the course of infection, the lipidome was analyzed between 0 and 7 DPI of infection. This analysis revealed distinct lipid dynamics in response to SARS-CoV-2 infection (Figure 6F). RD animals displayed a serum lipid shift in response to infection consisting primarily of decreased levels of phospholipids with mixed representation of lipid classes and a distribution of long chain and polyunsaturated fatty acids (PUFA). HFHS serum displayed a more drastic pattern of lipid depletion and enrichment. Specifically, HFHS serum reflected a sharp enrichment of free polyunsaturated fatty acids (PUFA) and a combination of enrichment and depletion of PUFA containing phospholipids. This response peaked at 7 DPI and began to return to homeostasis by 14 DPI, though certain lipid patterns were carried out until 21 DPI.

## 4. Discussion

The development of animal models that faithfully recapitulate certain aspects of human disease remains a top priority in SARS-CoV-2 research. Healthy Syrian hamsters develop mild to moderate disease like most human cases; however, they do not exhibit the more severe respiratory disease seen in humans with comorbidities such as obesity, diabetes, or other chronic illness [8,31,32]. Thus, we developed an experimental infection model of hamsters exclusively fed a high-fat high-sugar diet to model the impact of Western Diet on COVID-19 severity. In the Syrian hamster, this diet caused diet-induced morbidity, led to increased weight gain during adolescence, and ultimately led to an increased glucose tolerance, systemic hyperlipidemia, increased total cholesterol, and a liver pathology reminiscent of a NAFLD-like phenotype. The lack of net weight gain in this model may present a means of decoupling liver associated pathologies such as NAFLD from obesity-associated disease more broadly. In humans, NAFLD is predominantly a consequence of obesity and frequently associated also with other comorbidities as well [33]. In the context of COVID-19, NAFLD is associated with increased hospitalization and disease severity [34].

The morbidity observed in the absence of infection in the HFHS group should be considered in future studies utilizing this model. In particular, this feature of the model may make survival-based studies difficult. Human clinical studies of COVID-19 are plagued by this same difficulty in quantifying the contribution of infection and the associated comorbidities to the eventual cause of death. If appropriately controlled for in this model the relative contribution to death from the infection and the comorbidities can be quantified. We observed that male hamsters on a HFHS diet demonstrated delayed lower and upper respiratory tract clearance after infection with SARS-CoV-2, which was accompanied by more severe disease presentation. Our data is in agreement with findings in mice, which have reported enhanced morbidity in aged and diabetic obese mice in a mouse-adapted SARS-CoV-2 model [35]. Conversely, we also observed increased weight loss, pathology, delayed lung recovery, and influx of immune cells into the lung in a subset of hamsters fed a regular diet as compared to what has been shown in younger animals [17,19]. This is likely due to the increased age of the animals used in this study [36]. Previously, lung function analysis after SARS-CoV-2 infection in a rodent model has only been demonstrated in ACE2 mice [37]. While not significantly different between the diet groups, we performed functional lung analysis for the first time in the Syrian hamster after SARS-CoV-2 infection and demonstrated that this model also recapitulates increased total airway resistance and decreased inspiratory capacity. This suggests that the Syrian hamster, besides recapitulating lung pathology, may also be a useful model for mechanistic studies of the respiratory parameters affected by COVID-19.

Importantly, the HFHS Syrian hamster model presented here recapitulated two key mediators of severe human COVID-19, but only marginally. One unique feature of the cytokine profile in human disease is the elevation of IL-6 and IL-10, which have been indicated as causes of increased pathology [38,39,40,41]. In line with this, in HFHS animals we observed increases in serum IL-10 and IL-6 levels after infection, but these were not significant and more in-depth characterization is required to establish if the cytokine response in serum truly is affected. Secondly, in response to infection, HFHS animals showed a more severe response in their serum lipids at 7 DPI compared to RD animals. The lipids that dominated this response were free-PUFAs and PUFA-containing phosphatidylethanolamine (PE). In addition, we saw mixed increase and decrease of PUFA-containing plasmalogens and triacylglycerols. The metabolic comorbidities associated with severe COVID-19 were previously shown to correlate with specific mobilization of serum lipids in a human cohort [42]. Specifically, disease severity, defined by ICU admittance, was shown to be associated with increased free PUFAs and PUFA-containing phosphatidylethanolamine, as well as a decrease of PUFA-containing phosphatidylcholine and plasmalogen, compared to non-ICU hospitalized patients. These imbalances were reflected in the circulating milieu of immune-active, PUFA-derived lipid mediators in these patients. The lipid pattern findings in the Syrian hamster model suggest that these serum lipid changes are dependent on preexisting serum hyperlipidemia and stimulated by infection with SARS-CoV-2. Despite the lack of obesity in these animals, the matching of clinical SARS-CoV-2-associated lipid patterns and cytokine profile in this model supports its utility in examining lipid and inflammation dynamics associated immune dysregulation during infection.

Of note, this did not seem to adversely affect the humoral immune response while viral titers in oropharyngeal swabs and lung tissues suggested delayed clearance in the HFHS group. This may indicate that other immune pathways were disproportionately affected, but further investigations would be necessary to draw concrete conclusions.

This study had several limitations. Only male hamsters were used, which have been shown to show increased pathology upon infection with SARS-CoV-2 [43]. Additionally, only one age group was evaluated, and further studies are required to determine if there is an age-related bias with diet and disease severity. Finally, uninfected controls were not used in this study.

Taking the limitations of the model into account, our data further suggests the possible suitability of the Syrian hamster model to assess immunomodulatory therapies. While dietary advice for those suffering from metabolic diseases is proposed to reduce burden of severe COVID-19 [43], it remains doubtful if any change in diet can impact disease outcome favorably after infection has occurred. Targeted immunomodulatory therapies, such as anti-IL-6 therapies, may be more efficient [44]. The Syrian hamster model may also be applied to further studies of selected aspects of NAFLD, which the model recapitulates. This model seems to present with an absence or limited amount of liver fibrosis; further work is needed to demonstrate how faithfully it assesses the direct effect of liver fibrosis on acute disease. However, it may be useful to assess long term post-COVID-19 NAFLD, to document further deterioration of liver damage [45] and the relation to infection sequelae.

## Figures and Tables

**Figure 1 viruses-13-02506-f001:**
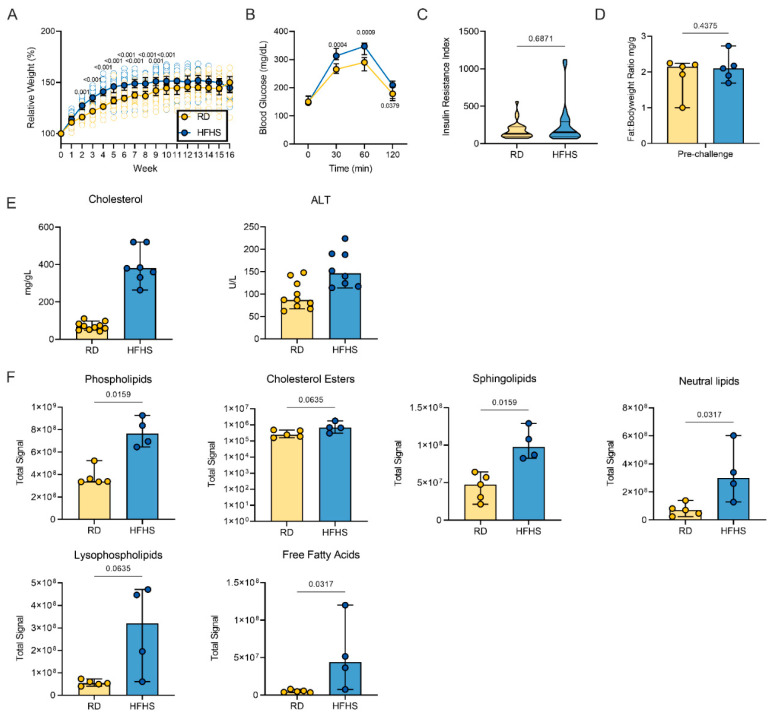
High-fat and high-sugar diet induces metabolic changes characterized by increased juvenile weight gain and glucose tolerance. Male Syrian hamsters were fed either a regular or high-fat high-sugar diet ad libitum for 16 weeks. (**A**) Relative weight gain in hamsters on each diet regimen, measured weekly. Graphs show median ± 95% CI, *n* = 35, ordinary two-way ANOVA, followed by Sidak’s multiple comparisons test. (**B**) Oral glucose tolerance test performed at 16 weeks. Graphs show median ± 95% CI, *n* = 30 (RD)/29 (HFHS), ordinary two-way ANOVA, followed by Sidak’s multiple comparisons test. (**C**) Insulin response after application of oral glucose load as shown by insulin resistance index (fasting glucose level (mmol/L) x fasting insulin level (mIU/L)). Truncated violin plots depicting median, quartiles and individuals, *n* = 30 (RD)/29 (HFHS), Mann–Whitney test. (**D**) Adiposity index as measured by testicular fat pads/total body weight at 16 weeks. Bar chart depicting median, 95% CI and individuals, *n* = 5, Mann–Whitney test. (**E**) Blood lipid ALT and cholesterol levels measured on a commercially available lipid panel on an automated blood chemistry analyzer *n* = 10 (RD)/8 (HFHS) (see Appendix A). (**F**) Serum aggregate lipids signal analyzed by liquid chromatography tandem mass spectrometry (LC-MS/MS) at 16 weeks of diet regimen. Bar chart depicting median, 95% CI and individuals, *n* = 5 (RD)/4 (HFHS), Mann–Whitney test. Abbreviations: RD = regular diet; HFHS = high-fat high-sugar; ALT = alanine aminotransaminase. *p*-values are indicated were appropriate.

**Figure 2 viruses-13-02506-f002:**
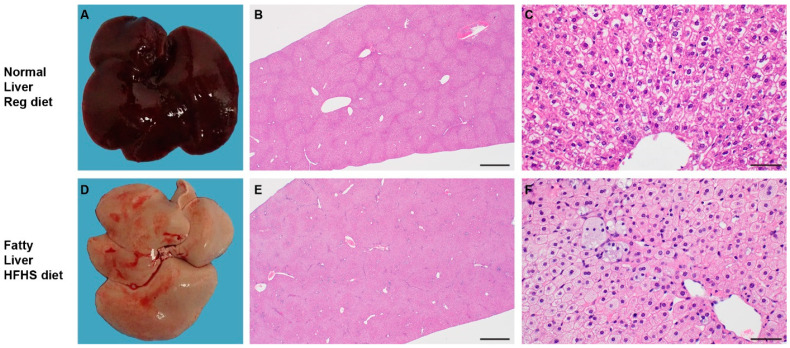
High-fat and high-sugar diet induces liver damage and systemic hyperlipidemia. Male Syrian hamsters were fed either a regular or high-fat high-sugar diet ad libitum and five animals from each group were sacrificed week 16 for analyses of liver tissue. (**A**,**D**) Gross imaging of a representative liver from one hamster on the RD and one hamster on the HFHS diet regimen. (**B**,**E**) 20× photomicrograph of H&E-stained slide. (**C**,**F**) 400× photomicrograph of H&E-stained slide. (**G**) RNA was isolated for gene expression analyses from liver tissue at 16 weeks. Using Integrated Pathway Analysis (Qiagen), significantly upregulated canonical pathways were identified. Graphs show pathways associated with cell recruitment, activation, and immunological inflammation (*p* > 0.05, *z*-score ≤ −2 or ≥ 2). (**H**) Integrated Pathway Analysis (Qiagen) was used to depict the gene network associated with nonalcoholic steatohepatitis. Symbols refer to legend below figure. Red: gene upregulation in high-fat high-sugar animals as compared to regular diet animals. Green: downregulation in comparison to regular diet.

**Figure 3 viruses-13-02506-f003:**
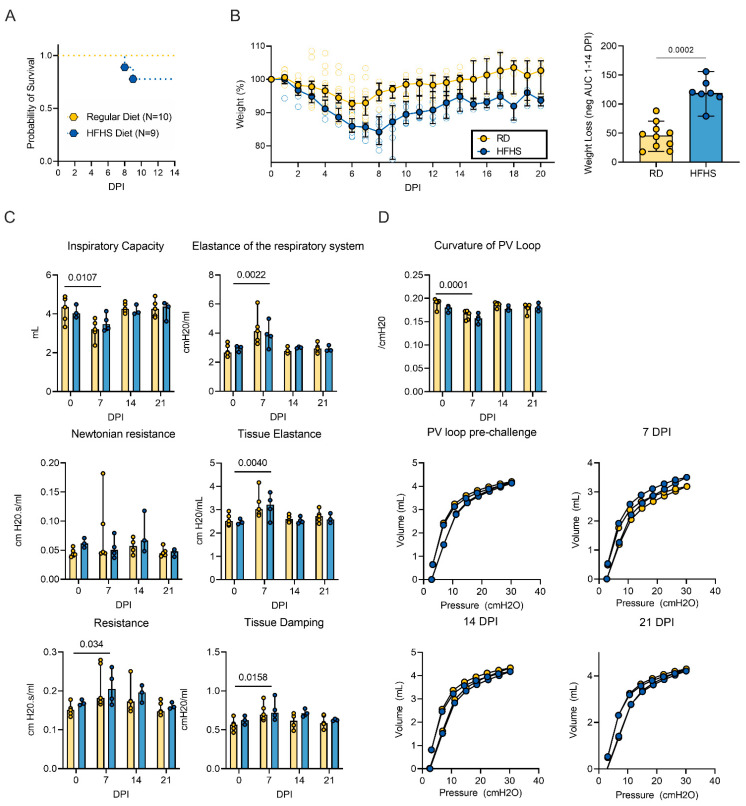
High-fat and high-sugar diet exasperated disease severity after SARS-COV-2 infection. Male Syrian hamsters were fed either a regular or high-fat high-sugar diet ad libitum for 16 weeks, then challenged with 8 × 10^4^ TCID50 SARS-CoV-2. (**A**) Survival after challenge for RD (*n* = 10) and HFHS (*n* = 9) in the 14 and 21 DPI groups. (**B**) Relative weight loss in hamsters after challenge. Left graph shows median ± 95% CI. Right graph shows area under the curve (AUC, negative peaks only) between 1 and 14 DPI of surviving animals. Bar chart depicting median, 95% 0 CI and individuals, *n* = 10 (RD)/7 (HFHS), Mann–Whitney test. (**C**) Lung function analysis after challenge. (**D**) Pressure–volume loops at pre-challenge, 7, 14, and 21 DPI. Abbreviations: RD = regular diet; HFHS = high-fat high-sugar; DPI = days post inoculation. *p*-values are indicated were appropriate.

**Figure 4 viruses-13-02506-f004:**
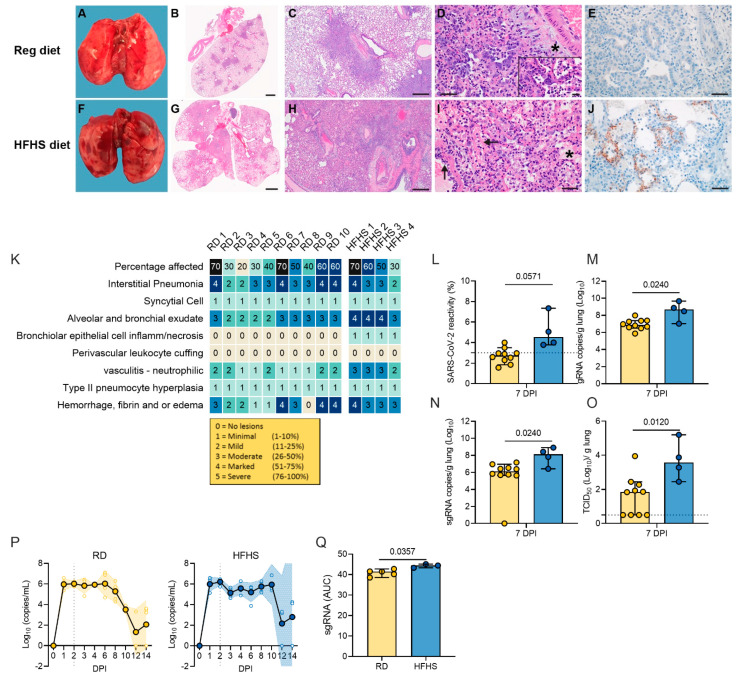
High-fat and high-sugar diet is associated to increased pulmonary pathology and decreased viral clearance. Animals were euthanized at 7 DPI with SARS-CoV-2 in order to compare lung pathology and viral titers. (**A**–**J**) Gross and photomicrographic images of hamster lungs taken at 7 DPI. (**A**,**F**) Gross necropsy findings consisted of multifocal well-circumscribed dark red foci throughout turgid lobes which failed to collapse. (**B**,**G**) Dark red foci in the gross images correlate with the consolidated foci adjacent to airways and scattered along the pleural margin in the sub-gross images. HE 1.4×. (**C**,**H**) Foci of interstitial pneumonia adjacent to terminal bronchioles and accompanying blood vessels. HE 20×. (**D**,**I**) Pneumonia consists of alveoli containing neutrophils, eosinophils, alveolar and septal macrophages, fibrin, edema and septa lined by hyperplastic type II pneumocytes, HE 400×. Syncytial cells are common (see inset, HE, 1000×). Pneumonic areas in the HFHS diet hamsters frequently had abundant intra-alveolar edema (*) and organizing fibrin (>) mixed with inflammatory cells. Note the vessel wall disrupted by sub-endothelial leukocytes and cellular debris (⇓). (**E**,**J**) Anti-SARS-CoV-2 immunoreactivity in the lungs from the regular diet hamsters is rare compared to the frequent pneumocyte immunoreactivity in the lungs of the HFHS diet hamsters, IHC, 400×. (**K**) Individual pathological scores. (**L**) Quantitative count of SARS-CoV-2 immunoreactivity by morphometric analysis. Bar chart depicting median, 95% CI, and individuals, *n* = 10 (RD)/4 (HFHS), Mann–Whitney test. (**M**,**N**) Lung viral load measured by g and sgRNA. Bar chart depicting median, 95% CI and individuals, *n* = 10 (RD)/4 (HFHS), Mann–Whitney test. (**O**) Infectious virus measured by lung titration. Bar chart depicting median, 95% CI and individuals, *n* = 10 (RD)/4 (HFHS), Mann–Whitney test. (**P**) Viral load in oropharyngeal swabs measured in sgRNA copy number for RD and HFHS animals. Graphs show median, individual animals and 95% CI (shaded area). Dotted line = peak shedding. (**Q**) Area under the curve (AUC) analysis of virus shedding shown in (**P**). Bar chart depicting median, 95% CI and individuals, 21 DPI: *n* = 5 (RD)/3 (HFHS), Mann–Whitney test. (**H**) Dotted line = limit of detection. Abbreviations: g = genomic; sg = subgenomic; DPI = days post inoculation; H&E = hematoxylin and eosin stain; IHC = immunohistochemistry. *p*-values are indicated were appropriate.

**Figure 5 viruses-13-02506-f005:**
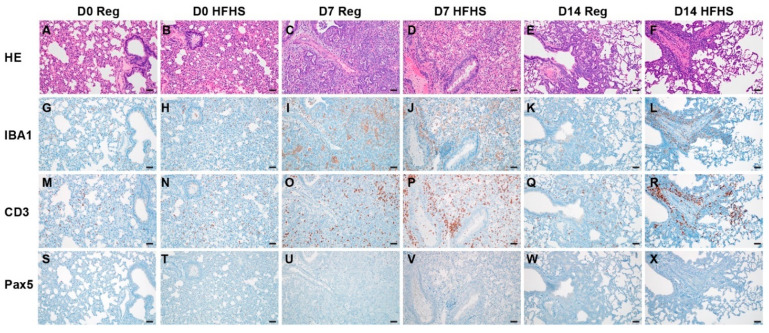
Immune infiltration and in the lung during acute-phase of infection and humoral immunity is not significantly affected by high-fat high-sugar diet. Animals were euthanized at 0, 7, and 14 DPI and the presence of SARS-CoV-2 antigen, T-cells, B-cells and macrophages investigated. (**A**,**B**) Pre-challenge RD and HFHS diet hamster lungs. (**G**,**H**) IBA1; (**M**,**N**) CD3 and (**S**,**T**) Pax5. (**C**,**D**) Lungs at 7 DPI. (**I**,**J**) IBA1; (**O**,**P**) CD3 and (**U**,**V**) Pax 5. (**E**,**F**) Lungs at 14 DPI. (**K**,**L**) IBA1; (**Q**,**R**) CD3 and (**W**,**X**) Pax 5. (**A**–**F**) HE. All images 200x. Abbreviations: RD = regular diet; HFHS = high-fat high-sugar; DPI = days post inoculation.

**Figure 6 viruses-13-02506-f006:**
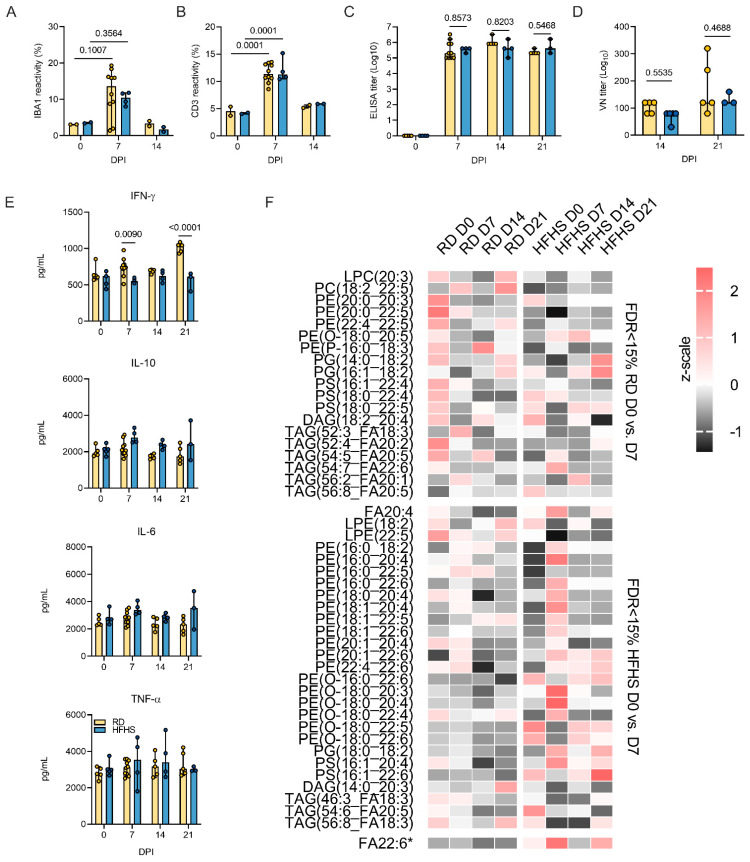
Disease manifestation is accompanied by prolonged viral shedding, systemic immune and metabolomic dysregulation after high-fat high-sugar diet. Animals were euthanized pre-challenge, at 7, 14, and 21 DPI with SARS-CoV-2 and serum and lung tissue collected for immune and lipid mediator analyses. (**A**,**B**) Lung infiltration of T-cells (CD3) and macrophages (IBA1) was quantified by morphometric analysis. Bar chart depicting median, 95% CI and individuals, pre-challenge and 14 DPI: *n* = 2, 7 DPI: *n* = 10 (RD)/4 (HFHS), ordinary two-way ANOVA, followed by Tukey’s multiple comparisons test. (**C**) ELISA titers against spike protein of SARS-CoV-2 (lineage A) in serum obtained pre-challenge, at 7, 14, and 21 DPI. Bar chart depicting median, 95% CI and individuals, pre-challenge and 14 DPI: *n* = 5 (RD)/4 (HFHS), 7 DPI: *n* = 10 (RD)/4 (HFHS), 21 DPI: *n* = 5 (RD)/3 (HFHS), ordinary two-way ANOVA, followed by Sidak’s multiple comparisons test. (**D**) Virus neutralization titers against SARS-CoV-2 (lineage A) in serum obtained at 14 and 21 DPI. Bar chart depicting median, 95% CI s and individuals, 14 DPI: *n* = 5 (RD)/4 (HFHS), 21 DPI: *n* = 5 (RD)/3 (HFHS), ordinary two-way ANOVA, followed by Sidak’s multiple comparisons test. (**E**) Serum levels (pg/mL) of INF-γ, TNFα-, IL-6, and IL-10 measured by ELISA from serum collected on 0, 7, 14, and 21 DPI. Bar chart depicting median, 95% CI and individuals, pre-challenge/14 and 21 DPI: *n* = 5 (RD)/4 (HFHS), 7 DPI: *n* = 10 (RD)/4 (HFHS), ordinary two-way ANOVA, followed by Sidak’s multiple comparisons test. (**F**) Lipid time-course heatmap: changes in PUFA-containing serum lipids associated with an active SARS-CoV-2 infection as measured by LC-MS/MS. Autoscaled intensities are displayed for serum lipid species that were significantly changed between 0 and 7 DPI in either regular diet or HFHS diet hamsters with a false discovery rate of 15% equating to *p* = 0.0256, 0.0193 for RD and HFHS, respectively. * FA22:6 (HFHS *p* = 0.0374) is displayed for comparison to clinical data despite not passing FDR filters. Abbreviations: TNF = tumor necrosis factor; IFN = interferon; IL = interleukin; RD = regular diet; HFHS = high-fat high-sugar; DPI = days post inoculation; = virus neutralization. *p*-values are indicated where appropriate.

## Data Availability

The data generated in this study have been deposited in the figshare repository 10.6084/m9.figshare.14840583. RNA-Seq data was deposited in NCBI GEO under accession number GSE175943 GEO Accession viewer (nih.gov).

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
