# Peer review of "High-Fat High-Sugar Diet-Induced Changes in the Lipid Metabolism Are Associated with Mildly Increased COVID-19 Severity and Delayed Recovery in the Syrian Hamster"

_viruses, 2021, doi:10.3390/v13122506_

Round 1
Reviewer 1 Report
Port/Adney et al present a well-written paper that shows that a high-fat, high-sugar diet is a slight risk factor for disease severity in SARS-CoV-2-infected Syrian hamsters. Overall, the methodology seems sound, the data set is very extensive, and the presentation of the diet-based hamster model is novel. The diet-related effects on disease severity are somewhat moderate but interesting.
I have the following comments:
- Throughout the paper, the authors refer to median values, whereas the mean values are often depicted. Given the low number of replicates, it is more informative to use the mean value.
- The authors repeatedly use violin plots to depict their data, even when the number of replicates is very low (n≤4). Although violin plots can be useful when depicting large Gaussian or multimodal distributions, violin plots are generally unsuitable in this publication because the type of distribution cannot be inferred from the data that is provided. Furthermore, sometimes data points are difficult to distinguish from the contours of the violin plots. It would be more useful to visualize the data points alongside the mean-value ± 95% confidence interval (even if the differences are not apparently significant).
- Age is a very relevant factor for disease severity mostly in combination with diet. As the authors used one age group of hamsters, please discuss or mention as limitation that there might be an age-related bias.
- In line 280 the authors mention an “incompatibility with the instrument”. Does this mean that the instrument was not suitable to detect hamster-specific HDL, LDL, please explain?
- It is redundant to refer to “COVID-19 disease severity” as disease is already a part of the abbreviation. The title should be changed to “COVID-19 severity.”
- 2G would benefit from either combining all three graphs into a single graph or giving them each an x-axis title to make things more clear for the reader.
- Poor image quality makes Fig. 2H unreadable.
- 4 Some of the x/y axes are difficult to read (too small).
- 2, 4, 5 please include scale bars in the pictures.
- Typos: Line 215 INF instead of IFN; Line 467 (“Turkey’s” to “Tukey’s”), Line 598 (“absences” to “absence”)
Author Response
Port/Adney et al present a well-written paper that shows that a high-fat, high-sugar diet is a slight risk factor for disease severity in SARS-CoV-2-infected Syrian hamsters. Overall, the methodology seems sound, the data set is very extensive, and the presentation of the diet-based hamster model is novel. The diet-related effects on disease severity are somewhat moderate but interesting.
I have the following comments:
- Throughout the paper, the authors refer to median values, whereas the mean values are often depicted. Given the low number of replicates, it is more informative to use the mean value.
We agree with the reviewer that it is important to have figures and text reflect each other, and we ensure that is the case. However, we understand that for the limited number of individuals, we cannot assume normal distribution, hence showing the median and 95% CI should be more accurate. We have ensured that in both figures and text we show the median unless otherwise stated and explained. Values for blood chemistry values and respiratory parameters should remain as medians. In these situations, the means can show too much variation, making the median the most correct value to report. (Lott JA, Smith DA, Mitchell LC, and Moeschberger ML. Use of medians and “averages of normals” of patients’ data for assessment of long-term analytical stability. Clin Chem. 1996 Jun;42( 6 Pt 1): 888-92.)
- The authors repeatedly use violin plots to depict their data, even when the number of replicates is very low (n≤4). Although violin plots can be useful when depicting large Gaussian or multimodal distributions, violin plots are generally unsuitable in this publication because the type of distribution cannot be inferred from the data that is provided. Furthermore, sometimes data points are difficult to distinguish from the contours of the violin plots. It would be more useful to visualize the data points alongside the mean-value ± 95% confidence interval (even if the differences are not apparently significant).
We thank the reviewer for their comment. We are happy to change the violin plots into bar charts also showing the individuals and median with 95% CI. We have removed statistic p-values from the AST and Cholesterol measurements, as these are indeed not appropriate.
- Age is a very relevant factor for disease severity mostly in combination with diet. As the authors used one age group of hamsters, please discuss or mention as limitation that there might be an age-related bias.
We have added a paragraph that discusses the limitations of this study.
Line 608: “This study had several limitations. Only male hamsters were used, which have been shown to show increased pathology upon infection with SARS-CoV-2 [44]. Additionally, only one age group was evaluated, and further studies are required to determine if there is an age-related bias with diet and disease severity. Finally, uninfected controls were not used in this study. “
- In line 280 the authors mention an “incompatibility with the instrument”. Does this mean that the instrument was not suitable to detect hamster-specific HDL, LDL, please explain?
Some values were not calculated because the machine could not recognize hamster specific HDL or LDL, but because the lipid concentrations were too high for the machine to calculate. We have corrected the text to make this less confusing for readers.
- It is redundant to refer to “COVID-19 disease severity” as disease is already a part of the abbreviation. The title should be changed to “COVID-19 severity.”
We thank the reviewer for this comment and have corrected the title accordingly.
- 2G would benefit from either combining all three graphs into a single graph or giving them each an x-axis title to make things more clear for the reader.
We have changed this to include x-axis titles in all three
- Poor image quality makes Fig. 2H unreadable.
We have addressed this.
- 4 Some of the x/y axes are difficult to read (too small).
We have addressed this. We have increased the figure sizes.
- 2, 4, 5 please include scale bars in the pictures.
We have added in scale bars.
- Typos: Line 215 INF instead of IFN; Line 467 (“Turkey’s” to “Tukey’s”), Line 598 (“absences” to “absence”)
We thank the reviewer for this comment and have corrected the title accordingly.

Reviewer 2 Report
Article: „High-fat high-sugar diet-induced changes in the lipid metabolism are associated with mildly increased COVID-19 disease severity and delayed recovery in the Syrian hamster“ by Port et al.
Summary: In their article, Port and colleagues show that a Western diet has exacerbating effect on the course of COVID 19 in Syrian hamsters. They use different state-of-the-art techniques and methods and convincingly demonstrate that a high-fat high-sugar diet results in increased lung pathology and prolonged viral shedding. The paper is conclusive, nicely written and should have an immediate impact on COVID-19 research which is why it should be published a.s.a.p. I am listing a few points that should be addressed by the authors while preparing a revised version for publication. Focus should be set on cautiousness and thoroughness when preparing revised figures and figure legends!
Major:
- None.
Minor:
- Your reference # 6 (Obesity and mortality of COVID-19. Meta-analysis) has been retracted. You should replace that reference.
- Which passage of nCoV-WA1-2020 was used for in vivo infection experiments? (L69f.)
- In total, how many hamsters did you use for your experiments? (L76f.)
- L125f: How did you determine sample quality from the information presented in Table S1 (Liver marker profile in serum of regular diet (RD) and high-fat high-sugar diet (HFHS) after 16 weeks). Please clarify.
- L127f: Why did you sequence the virus stocks? Just to confirm that the virus you got from the CDC (nCoV-WA1-2020, which apparently already has an accession #) is correct?
- Why did you choose to include only male hamsters in your study?
- L158: “McGovern TK JOVE 2013” should be a proper reference … see http://dx.doi.org/10.3791/50172.
- L241 and elsewhere: I think it’s okay to only show the p values in the figure and not in the text as bulky insertions to improve the read flow.
- Fig. 1A: Somewhere, you should discuss that the relative weight gain in HFSF hamsters is only transient when compared to RD.
- Fig 1E: N=xxx is missing. Seems like you tested 10/7 animals (cholesterol) vs. 10/8 animals (ALT), any explanation for that?
- Fig. 2H is barely readable. Please increase the size of this panel as well as font sizes and resolution.
- L326: “8x104” should be “8x104”.
- Fig. 3A: “Regukar Diet” needs to be corrected.
- There were attempts to standardize histopathologic lesions in lungs of SARS-CoV-2 infected hamsters. Did you follow any of these “guidelines”? Please discuss.
- L474: There’s no dotted line in E… The panels and the figure legend don’t fit each other: There’s no panel G, H or I in the figure (contrary to what’s stated in the legend).
- With “antiviral interferon”, do you refer to IFN gamma? (L500f.)
- L529f: Some dwarf hamsters seem to develop pretty robust severe COVID-19. Would your approach be transferable to other hamster species?
- L535: an increased (typo)!
- Briefly discuss the limitation that uninfected control animals were not included in your study.
- Fig. S2: Colors are missing in the legend (RD, HFHS).
- Figs S3, S4 and S5 as well as Fig. 2B, C, E, F, Fig 4B-E and G-J, and Fig. 5: add scale bars.
- Tab. S1: “~~ +” should be “~~~“, correct?
Author Response
Article: „High-fat high-sugar diet-induced changes in the lipid metabolism are associated with mildly increased COVID-19 disease severity and delayed recovery in the Syrian hamster“ by Port et al.
Summary: In their article, Port and colleagues show that a Western diet has exacerbating effect on the course of COVID 19 in Syrian hamsters. They use different state-of-the-art techniques and methods and convincingly demonstrate that a high-fat high-sugar diet results in increased lung pathology and prolonged viral shedding. The paper is conclusive, nicely written and should have an immediate impact on COVID-19 research which is why it should be published a.s.a.p. I am listing a few points that should be addressed by the authors while preparing a revised version for publication. Focus should be set on cautiousness and thoroughness when preparing revised figures and figure legends!
- Major:
- - None.
- Minor:
- Your reference # 6 (Obesity and mortality of COVID-19. Meta-analysis) has been retracted. You should replace that reference.
We have replaced the reference and thank the reviewer for their comment.
- Which passage of nCoV-WA1-2020 was used for in vivo infection experiments? (L69f.)
This was passage 4; information has been added to the methods.
- In total, how many hamsters did you use for your experiments? (L76f.)
N = 70 for all data presented in this manuscript, we have added this to the methods.
- L125f: How did you determine sample quality from the information presented in Table S1 (Liver marker profile in serum of regular diet (RD) and high-fat high-sugar diet (HFHS) after 16 weeks). Please clarify.
This serum was run on a standard chemistry analyzer. As we could not validate each value in the panel, these were compared to previously published values from both healthy and obese hamsters and interpreted alongside supporting data.
- L127f: Why did you sequence the virus stocks? Just to confirm that the virus you got from the CDC (nCoV-WA1-2020, which apparently already has an accession #) is correct?
All virus stocks in our laboratory are sequenced before use in experiments. Propagation of stocks can result in changes of the viral sequence, loss of cleavage sites etc. To ensure we are still using the virus as received and represented by the accession #) we find it prudent to sequence after we have grown a stock in house.
- Why did you choose to include only male hamsters in your study?
Although there are clearly sex-differences in pathogenesis after infection with any agent, we elected to use only male hamsters after reports that they develop more severe pathology after infection with SARS-CoV-2. We believe that further studies would be interesting to better tease out the impacts of sex in this model and have added a statement describing this limitation.
- L158: “McGovern TK JOVE 2013” should be a proper reference … see http://dx.doi.org/10.3791/50172.
This has been added.
- L241 and elsewhere: I think it’s okay to only show the p values in the figure and not in the text as bulky insertions to improve the read flow.
While we agree with the reviewer that it may impede the flow, we do find it relevant to also state in the text which values were compared and/or the p value. We respectfully suggest leaving them in the text.
- Fig. 1A: Somewhere, you should discuss that the relative weight gain in HFSF hamsters is only transient when compared to RD.
We have added language describing this transient difference.
Line 243: “Initially, animals on the HFHS diet gained weight faster than animals on the regular diet, although this was a transient difference.”
- Fig 1E: N=xxx is missing. Seems like you tested 10/7 animals (cholesterol) vs. 10/8 animals (ALT), any explanation for that?
This indeed correct and we have added the N to the legend. One animal in the HFHS group had a chol value which could not be calculated as the conc was too high. This is further shown in S Table 1.
- Fig. 2H is barely readable. Please increase the size of this panel as well as font sizes and resolution.
This has been changed.
- L326: “8x104” should be “8x104”.
.This has been corrected.
- Fig. 3A: “Regukar Diet” needs to be corrected.
We thank the reviewer for their attention to detail, this is highly appreciated. We have corrected this.
- There were attempts to standardize histopathologic lesions in lungs of SARS-CoV-2 infected hamsters. Did you follow any of these “guidelines”? Please discuss.
The pathological assessment was performed by a board-certified pathologist. They used the same or similar criteria which other pathologists were using to publish on the Syrian hamster SARS2 model, including published studies from our laboratory and institute. We have added a sentence in the manuscript commenting on this.
Line 180: Histopathology was assessed by a board-certified veterinary pathologist using criteria as previously applied to the Syrian hamster SARS CoV-2 model.”
- L474: There’s no dotted line in E… The panels and the figure legend don’t fit each other: There’s no panel G, H or I in the figure (contrary to what’s stated in the legend).
This has been corrected and the legends of Figure 4 and 5 changed accordingly.
- With “antiviral interferon”, do you refer to IFN gamma? (L500f.)
Formatting must have taken issue with Greek letters. This is correct and we have changed it in the manuscript.
- L529f: Some dwarf hamsters seem to develop pretty robust severe COVID-19. Would your approach be transferable to other hamster species?
This seems possible, however, there is no data so far on using a HFHS diet in dwarf hamsters as far as we are aware. It seems that diabetes and obesity are issues observed in pets, however, which suggest that such a diet can be also applied in this species.
- L535: an increased (typo)!
We thank the reviewer for this comment and have corrected this.
- Briefly discuss the limitation that uninfected control animals were not included in your study.
We thank the reviewer for this comment. We have added this to our paragraph describing the study limitations. Line 608: “This study had several limitations. Only male hamsters were used, which have been shown to show increased pathology upon infection with SARS-CoV-2 [44]. Additionally, only one age group was evaluated, and further studies are required to determine if there is an age-related bias with diet and disease severity. Finally, uninfected controls were not used in this study. “
- Fig. S2: Colors are missing in the legend (RD, HFHS).
We thank the reviewer for noticing the figure legend has been corrected.
- Figs S3, S4 and S5 as well as Fig. 2B, C, E, F, Fig 4B-E and G-J, and Fig. 5: add scale bars.
These have been added.
- Tab. S1: “~~ +” should be “~~~“, correct?
Correct, we thank you for the comment and have corrected this.
